# A Pilot Study on the Effect of Anti-Thrombopoietin Antibody on Platelet Count in Patients with Type 2 Diabetes

**DOI:** 10.3390/molecules25071667

**Published:** 2020-04-04

**Authors:** Takuya Fukuda, Masahide Hamaguchi, Takafumi Osaka, Yoshitaka Hashimoto, Emi Ushigome, Mai Asano, Masahiro Yamazaki, Eriko Fukuda, Kei Yamaguchi, Koji Ogawa, Naoki Goshima, Michiaki Fukui

**Affiliations:** 1Department of Endocrinology and Metabolism, Kyoto Prefectural University of Medicine, Graduate School of Medical Science, Kyoto 602-8566, Japan; takuya07@koto.kpu-m.ac.jp (T.F.); tak-1314@koto.kpu-m.ac.jp (T.O.); y-hashi@koto.kpu-m.ac.jp (Y.H.); emis@koto.kpu-m.ac.jp (E.U.); maias@koto.kpu-m.ac.jp (M.A.); masahiro@koto.kpu-m.ac.jp (M.Y.); michiaki@koto.kpu-m.ac.jp (M.F.); 2Department of Endocrinology and Diabetology, Ayabe City Hospital, Ayabe 623-0011, Japan; 3Molecular Profiling Research Center for Drug Discovery, National Institute of Advanced Industrial Science and Technology, Tokyo 135-0064, Japan; eriko-fukuda@aist.go.jp (E.F.); kei-yamaguchi@aist.go.jp (K.Y.); n-goshima@aist.go.jp (N.G.); 4ProteoBridge Corporation, Tokyo 135-0064, Japan; ogawa@proteo-bridge.co.jp

**Keywords:** autoantibody, cross-sectional study, diabetes, platelet

## Abstract

Thrombopoietin (THPO) is a circulatory cytokine that plays an important role in platelet production. The presence of anti-THPO antibody relates to thrombocytopenia and is rarely seen in hematopoietic and autoimmune diseases. To date, there had been no reports that focused on the anti-THPO antibody in patients with type 2 diabetes mellitus (T2DM). To evaluate prevalence of the anti-THPO antibody in patients with T2DM and the relationship between anti-THPO antibody and platelet count, a cross-sectional study was performed on 82 patients with T2DM. The anti-THPO antibody was measured by ELISA using preserved sera and detected in 13 patients. The average platelet count was significantly lower in patients with the anti-THPO antibody than in those without the anti-THPO antibody. Multivariate linear regression analyses showed a significant relationship between the anti-THPO antibody and platelet count, after adjusting for other variables. To our best knowledge, this was the first report on the effect of the anti-THPO antibody on platelet count in patients with T2DM. Further investigation is needed to validate the prevalence and pathological significance of the anti-THPO antibody in patients with T2DM.

## 1. Introduction

Thrombopoietin (THPO) is a circulatory cytokine that is produced mainly by the liver [1] and plays an important role in the proliferation and differentiation of megakaryocyte progenitors [2]. Proplatelet processes are formed from megakaryocytes, which later fragment into platelets [3]. The importance of THPO signaling in maintaining the number of circulating platelets had been reported, based on the reduced number of platelets and megakaryocytes after administration of the anti-THPO antibody in THPO knockout mice [4] and in mice with the THPO knockout-like phenotype [5]. In clinical practice, recombinant human THPO (rhTHPO) had been used for patients with severe thrombocytopenia [6,7].

The anti-THPO antibody was reported to be found in patients with amegakaryocytic thrombocytopenic purpura [8], idiopathic thrombocytopenia purpura (ITP) [9], systemic lupus erythematosus (SLE) [10,11], in those treated with rhTHPO [12,13] and recombinant human erythropoietin [14]. In those reports, the anti-THPO antibody was assumed to be the cause of thrombocytopenia. Shin SK et al. [14] reported the inhibitory activity of the anti-THPO antibody against the interaction between rhTHPO and the THPO receptor. They also reported that, using the BaF3 cell expression of the THPO receptor, a serum was able to partially neutralize the proliferative activity of rhTHPO. On the other hand, development of the anti-THPO antibody in healthy subjects has not been described [11].

Diabetes mellitus (DM) had been considered a prothrombotic state, in association with hyperglycemia, insulin resistance, and inflammation [15]. Patients with DM, particularly those with type 2 DM (T2DM), had been shown to exhibit increased platelet reactivity [16]. In addition to that, platelet count is a significant clinical variable used in markers of liver fibrosis such as the fibrosis 4 (FIB4) index [17] in the field of diabetes care, because patients with T2DM had been known to have a high prevalence of fatty liver disease, which spans a wide spectrum from simple steatosis to cirrhosis [18]. Therefore, the use of platelet indices, including platelet count, has recently attracted attention for patients with T2DM [19,20].

Although several factors that affect platelet count had been reported [21,22], a hidden factor that affects the platelet count may be present in patients with T2DM. In this pilot study, we reported that the anti-THPO antibody was relatively high in patients with T2DM and suggested the possible relationship between the anti-THPO antibody and platelet count.

## 2. Results

Between January 2015 and August 2016, we enrolled 181 patients (96 males and 85 females) in the cohort study. For the extraction of patients for analysis, we excluded patients who met the exclusion criteria. Consequently, a total of 82 patients (35 males and 47 females) were included in the final analysis (Figure 1). All patients had no history of hematologic diseases and receipt of rhTHPO. None of the patients had liver cirrhosis or enlarged spleen on abdominal ultrasonography. The distinct clinical features of autoimmune diseases, including type 1 DM and SLE were not evident in any of the patients.

The clinical and laboratory characteristics of the patients according to the presence or absence of the anti-THPO antibody are summarized in Table 1. The anti-THPO antibody was found in the preserved sera of 13 patients, and there were no significant differences in age and sex between patients with the anti-THPO antibody and those without the anti-THPO antibody. The average platelet count was significantly lower in patients with the anti-THPO antibody than in those without the anti-THPO antibody (Figure 2). The platelet count was less than 150 × 10^9^/L in four of thirteen patients with the anti-THPO antibody and in three of sixty-eight patients without the anti-THPO antibody. The parameters of glycemic control and chronic complications of diabetes were almost the same between the two groups.

Table 2 shows the correlation between platelet count and the clinical parameters. In the univariate correlation analyses, platelet count was positively correlated with HbA_1c_, CAP, and white blood cell count and was negatively correlated with age, MPV, and the presence of the anti-THPO antibody. Therefore, we selected age, HbA_1c_, CAP, white blood cell count, and MPV as covariates of multivariate linear regression analyses. Multivariate linear regression analyses showed a significant relationship between the anti-THPO antibody and platelet count, after adjusting for the other variables (β = −0.23; *p* < 0.05).

## 3. Discussion

To the best of our knowledge, this was the first report on the prevalence of the anti-THPO antibody and effect of the anti-THPO antibody on platelet count in patients with T2DM. Our pilot study showed that nearly 16% of patients with T2DM had the anti-THPO antibody. Although we could not demonstrate the causality between the presence of the anti-THPO antibody and decreased platelet count in this study, a previous report indicated the direct effect of the anti-THPO antibody on thrombocytopenia [14]; in that study, the anti-THPO antibodies might have blocked the normal function of thrombopoietin and resulted in decreased platelet production.

The platelets of patients with T2DM had been reported to be in a hyperreactive state, with exacerbated adhesion, aggregation, and activation [23]. Among the platelet indices, the MPV, which reflects platelet activity, was reported to be relevant in metabolic diseases, such as T2DM and fatty liver disease [16,24,25]. Among the pathologic spectrum of fatty liver disease, liver fibrosis had been the most related to platelet count reduction [26]. Therefore, we gave careful consideration on the influence of fatty liver disease on the platelet count by excluding patients with possible liver fibrosis, which was assessed by the LSM on transient elastography. In this study, the FIB4 index was higher in patients with the anti-TPHO antibody than in those without the anti-TPHO antibody; on the other hand, the LSM did not significantly differ between the two groups. It might have been overestimated in patients with the anti-THPO antibody because of the antibody-induced decreased number of platelets, although the FIB4 index is a useful noninvasive marker of fibrosis. Furthermore, when we see patients with low platelets and no findings of suggesting nonalcoholic steatohepatitis (NASH) or cirrhosis, the possibility of anti-THPO antibody needs to be considered, in addition to drug-related thrombocytopenia or ITP.

The reason for the production of such autoantibody in patients with T2DM remains unknown. Although we do not have clear explanations, there is one hypothesis that the anti-THPO is an autoantibody induced by chronic inflammation. Recently, the production of autoantibodies in patients with chronic inflammation has attracted attention. Misfolded proteins that form a complex with MHC class II molecules might be recognized as “neo-self” antigens by immune cells, which induce the production of autoantibodies [27]. Jagannathan-Bogdan M et al. [28] reported that patients with T2DM had T-cells that had increased production of IFN-γ and decreased percentage of regulatory T-cells that suppressed excessive immune activity. These reports indicated that the T-cells in patients with T2DM were skewed towards the proinflammatory T-helper subsets, which likely promoted chronic inflammation through elevated cytokine production.

This study had some limitations that warrant discussion. First, the ELISA assays for the anti-THPO antibody was a qualitative assessment; therefore, we were not able to verify the relationship between platelet count and anti-THPO antibody titers. Second, this study did not include healthy individuals; therefore, we were not able to compare the anti-THPO antibody between T2DM patients and healthy individuals. Third, bone marrow biopsy was not performed in all patients; therefore, we were not able to analyze the relationship between bone marrow megakaryocytes and the anti-THPO antibody or the other relevant clinical parameters. Fourth, platelet-associated IgG was not checked, because it was difficult to measure using frozen serum samples; therefore, we could not entirely exclude the possibility of ITP in the study population. Lastly, this study comprised Japanese patients only, and the findings might not be generalized to different racial and ethnic groups.

In conclusion, we reported the possible relationship between the anti-THPO antibody and platelet count in patients with T2DM. Unexplained mild thrombocytopenia in patients with T2DM might be due to an overlooked presence of the anti-THPO antibody. Further investigation is needed to validate the prevalence and pathological significance of the anti-THPO antibody in patients with T2DM.

## 4. Materials and Methods

### 4.1. Enrolled Patients and Anthropometry

We conducted a cross-sectional study on patients with DM who were registered in a cohort study, which aimed to obtain the clinical features of diabetic patients with fatty liver disease. The cohort study was examined and approved by the institutional review board (IRB) of Kyoto Prefectural University of Medicine (ERB-C-297-3) and the research protocol was registered at the UMIN (R000019143 and UMIN000035127). This study was performed in accordance with the Good Clinical Practice guidelines [29]. Before we enrolled a patient in the cohort study, the researcher or subresearcher explained the study design to each candidate patient using a brochure and an informed consent form. Written informed consent was given by each patient prior to study participation.

Blood test and abdominal ultrasonography with transient elastography were performed on all the recruited patients. We recorded the demographics, medical history, smoking status, amount of alcoholic beverages consumed per week, and medication usage. Body weight and height were measured in all the recruited patients. BMI was calculated by dividing the body weight in kilograms by the square of the height in meters.

We excluded patients due to the following criteria: Types of DM other than type 2; average alcohol consumption of >30 g/day for men and >20 g/day for women [30,31]; and patients with hepatitis B (positive for serum hepatitis B surface antigen), hepatitis C (positive for hepatitis C viral antibody), autoimmune hepatitis (positive for autoimmune serology), severe renal impairment (estimated glomerular filtration rate < 30 mL/min/1.73 m^2^), and history of malignant tumors and kidney transplantation. In addition, patients with findings of hepatic fibrosis on transient elastography (liver stiffness measurement (LSM) > 7 kPa) [32] were excluded.

### 4.2. Biochemical Assays

Venous samples were obtained in sterile standard tubes after >10 h of overnight fasting. Serum samples for enzyme-linked immunosorbent assay (ELISA) were stored at −80 °C until use. A central laboratory institute performed the blood tests. The platelet count, mean platelet volume (MPV), and other blood cell indices were analyzed by the Beckman Coulter method. Hemoglobin A_1c_ (HbA_1c_) was assayed using high-performance liquid chromatography. We calculated the estimated glomerular filtration rate (eGFR) using the Japanese Society of Nephrology equation—eGFR (mL/min/1.73 m^2^) = 194 × Cr^−1.094^ × Age^−0.287^ (×0.739 for females) [33]. The FIB4 index was calculated according to the formula as: Age (years) × aspartate aminotransferase (IU/L)/platelet count (×10^9^/L) × √alanine aminotransferase (IU/L) [17]. The serum THPO levels in the frozen samples were measured by a commercial sandwich ELISA (Quantikine; R&D Systems, Minneapolis, MN, USA), according to the manufacturer’s instructions.

### 4.3. Detection of Anti-THPO Antibodies

The anti-THPO antibody was detected by the custom-made ELISA. For the ELISA, microtiter plates were treated with a GSH solution (50 mM glutathione, 1 mM EDTA, 10 mM NaH_2_PO_4_, and 150 mM NaCl). The recombinant human THPO (HQ258196.1, N-term GST-tagged, 230 ng per well), which was expressed by a wheat germ cell-free system, was incubated in the wells for 1 h at 25 °C. Human IgG, which was expressed by a wheat germ cell-free system, was used as the positive control, and mock (H_2_O instead of mRNA) was used as the negative control. The ELISA plates made in this way were stored at −80 °C until use, then thawed at 25 °C before adding sera. After incubation for 1 h at 25 °C, the wells were treated with goat anti-human IgG ((H+L) Cross-Adsorbed Secondary Antibody HRP (1:10,000), A18811, Thermo Fisher Scientific, Waltham, MA, USA) for 1 h at 25 °C, and incubated with 100 µL of tetramethylbenzidine–H_2_O_2_ (Pierce™ TMB substrate kit, Thermo Fisher Scientific, Waltham, MA, USA). After 30 min at 25 °C, the reaction was stopped by the addition of 100 µL of 2 M H_2_SO_4_, and the yellow dye was measured at 450 nm. The threshold (average of 2 mock wells + 1.5× the difference of 2 mock wells) was calculated for each serum sample.

### 4.4. Transient Elastography

Measurements of liver stiffness and the controlled attenuation parameter (CAP) were performed using the transient elastography system (FibroScan^®^, Echosens, Paris, France) [34]. All patients underwent LSM using the 3.5-MHz standard M probe or the 2.0-MHz XL probe. The CAP had been designed to measure the liver ultrasound attenuation at 3.5 or 2.0 MHz, based on the signals acquired by the transient elastography system [35,36]. The CAP value was calculated only when the LSM was valid. The cut-off value of CAP was 237.8 dB/m for steatosis grade 1 (S1), based on a previous report [35], and we defined steatosis ≥ S1 as fatty liver [37].

### 4.5. Statistical Analysis

The JMP version 12.0 software (SAS Institute Inc., Cary, NC, USA) was used to perform all statistical analyses. The mean values were expressed as mean ± standard deviation (SD). All statistical tests were two-tailed and we considered a *p*-value of < 0.05 as statistically significant. Differences in categorical variables between the two groups were evaluated with the Fisher’s exact test. Continuous variables with normal distribution were compared using the Student’s *t*-test; continuous variables with non-normally distributed variables were compared by the Mann–Whitney *U*-test. Pearson’s correlation was used to verify the relationship between platelet count and the various clinical parameters. Multivariate linear regression analyses were used to evaluate the independent determinants of platelet count.

## Figures and Tables

**Figure 1 molecules-25-01667-f001:**
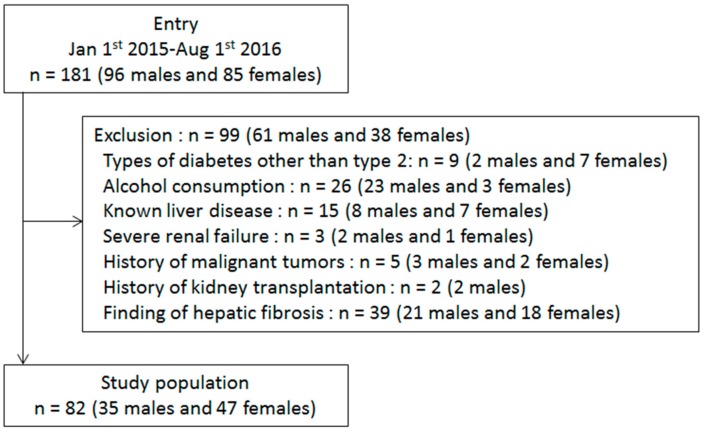
The study flow diagram.

**Figure 2 molecules-25-01667-f002:**
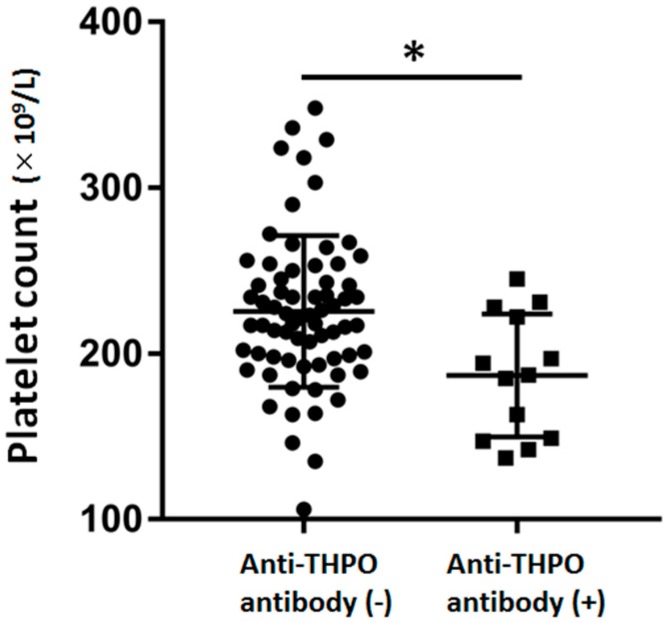
Comparison of the platelet count according to the presence/absence of the anti-THPO antibody. The data of the two groups were compared by the Student’s *t*-test (* *p*-value < 0.05). Lines indicate the mean and standard deviation for each group. THPO: Thrombopoietin.

**Table 1 molecules-25-01667-t001:** Clinical and laboratory characteristics of the patients (*n* = 82).

Characteristics	Anti-THPO Antibody (−)	Anti-THPO Antibody (+)	*p*
Total number	69	13	-
Male	30 (43.5)	5 (38.5)	0.74
Age (years)	64.1 ± 11.3	69.9 ± 10.2	0.09
Duration of diabetes (years)	13.9 ± 9.8	9.6 ± 7.7	0.15
Body weight (kg)	63.6 ± 13.7	57.3 ± 14.3	0.14
Body mass index (kg/m^2^)	24.3 ± 3.8	23.7 ± 4.1	0.56
Smoking (none/past/current)	40 (58.0)/23 (33.3)/6 (8.7)	4 (30.8)/8 (61.5)/1 (7.7)	0.15
Consumption of alcohol (g/day)	13.8 ± 4.7	8.8 ± 8.8	0.67
Retinopathy (NDR/SDR/PDR)	51 (73.9)/13 (18.8)/5 (7.3)	10 (76.9)/1 (7.7)/2 (1.4)	0.44
Nephropathy (normo/micro/macroalbumiuria)	45 (65.2)/19 (27.5)/5 (7.3)	10 (76.9)/3 (23.1)/0 (0)	0.74
Neuropathy (no/yes)	44 (63.8)/25 (36.2)	6 (46.2)/7 (53.8)	0.23
Sulfonylurea	26 (37.7)	6 (46.2)	0.57
Dipeptidyl peptidase-4 inhibitor	16 (23.2)	5 (38.5)	0.26
Glucagon-like peptide-1 receptor agonist	4 (5.8)	0 (0)	0.23
Pioglitazone	23 (33.3)	6 (46.2)	0.38
Metformin	32 (46.4)	4 (30.8)	0.29
SGLT2 inhibitor	2 (2.9)	0 (0)	0.40
Insulin therapy	22 (31.9)	4 (30.8)	0.94
Systolic blood pressure (mmHg)	130.1 ± 17.3	137.8 ± 22.3	0.16
Diastolic blood pressure (mmHg)	76.2 ± 11.8	81.0 ± 11.8	0.18
LSM (kPa)	4.63 ± 1.13	5.08 ± 1.29	0.19
CAP (dB/m)	262.1 ± 55.0	247.8 ± 55.9	0.39
Fatty liver	47 (68.1)	7 (53.8)	0.33
FIB4 index	1.51 ± 0.68	1.92 ± 0.58	<0.05
HbA_1c_ (%)	7.43 ± 1.04	7.44 ± 1.25	0.97
Fasting plasma glucose (mg/dL)	138.5 ± 35.5	144.6 ± 37.6	0.57
Aspartate aminotransferase (IU/L)	23.2 ± 9.4	25.2 ± 9.2	0.49
Alanine aminotransferase (IU/L)	23.9 ± 16.5	27.0 ± 16.7	0.53
Gamma-glutamyl transferase (IU/L)	32.8 ± 23.1	30.5 ± 17.3	0.73
eGFR (mL/min/1.73 m^2^)	76.6 ± 20.0	73.9 ± 19.1	0.66
C-reactive protein (mg/dL)	0.05 ± 0.03	0.14 ± 0.21	0.11
Thrombopoietin (pg/mL)	67.4 ± 63.0	91.1 ± 56.9	0.23
White blood cell count (×10^9^/L)	7.07 ± 1.83	5.60 ± 1.36	<0.01
Red blood cell count (×10^12^/L)	4.57 ± 0.50	4.73 ± 0.56	0.31
Hemoglobin (g/dL)	13.6 ± 1.5	14.1 ± 1.4	0.31
MPV (fL)	9.99 ± 0.91	9.68 ± 0.77	0.26
Platelets (×10^9^/L)	228.7 ± 53.3	186.7 ± 37.1	<0.01

Data are presented as n (%) or as mean ± SD. CAP: Controlled attenuation parameter; eGFR: Estimated glomerular filtration rate; FIB4: Fibrosis 4; HbA_1c_: Hemoglobin A_1c_; LSM: Liver stiffness measurement; MPV: Mean platelet volume; NDR: No diabetic retinopathy; PDR: Proliferative diabetic retinopathy; SDR: Simple diabetic retinopathy; SGLT2: Sodium-glucose cotransporter 2.

**Table 2 molecules-25-01667-t002:** Univariate correlation (left columns) and multivariate linear regression (right columns) determining the factors that affect platelet count (*n* = 82).

Characteristics	Univariate		Multivariate	
	*r*	*p*	*β*	*p*
Age (years)	−0.35	<0.01	−0.27	<0.01
Sex (male)	0.09	0.41	-	-
Duration of diabetes (years)	−0.09	0.42	-	-
HbA_1c_ (%)	0.32	<0.01	0.25	<0.05
eGFR (mL/min/1.73 m^2^)	0.21	0.06	-	-
C-reactive protein (mg/dL)	0.15	0.19	-	-
LSM (kPa)	0.15	0.17	-	-
CAP (dB/m)	0.25	<0.05	0.12	0.23
White blood cell count (×10^9^/l)	0.24	<0.05	0.14	0.18
Hemoglobin (g/dL)	0.03	0.76	-	-
MPV (fL)	−0.25	<0.05	−0.34	<0.01
Thrombopoietin (pg/mL)	0.02	0.84	-	-
anti-THPO antibody (yes)	−0.29	<0.01	−0.23	<0.05

r indicates the correlation coefficient and β indicates the multivariate linear regression coefficients. CAP: Controlled attenuation parameter; eGFR: Estimated glomerular filtration rate; HbA_1c_: Hemoglobin A_1c_; LSM: Liver stiffness measurement; MPV: Mean platelet volume; THPO: Thrombopoietin.

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
