# Peer review of "A Pilot Study on the Effect of Anti-Thrombopoietin Antibody on Platelet Count in Patients with Type 2 Diabetes"

_molecules, 2020, doi:10.3390/molecules25071667_

Round 1
Reviewer 1 Report
There are novelty, however remains some important concerns.
1) How is a nest case-control study? how to design and perform.
2) How about the baseline characteristics?
3) How to define confounding factors in multivariate linear regression?
Author Response
To Reviewer: 1
1) How is a nested case-control study? How to design and perform.
We thank you for your careful reading of the manuscript and helpful comment. As you felt out of place, we actually performed a cross-sectional study, not a nested case-control study. We have changed the phrase “nested case-control study” to “cross-sectional study” (Page 1, line 25; Page 1, line 34; Page 7, line 158).
Since January 2015, we performed a single center multipurpose cohort study in Japanese university hospital. The aim of the cohort study was to obtain the clinical features of diabetic patients with nonalcoholic fatty liver disease (NAFLD) and to prevent the progression of nonalcoholic steatohepatitis (NASH). For this purpose, we analyzed the clinical data registered in the study and found clinical features associated with the progression of NASH.
In this time, we enrolled patients participated in the cohort between January 2015 and August 2016. For the extraction of patients for analysis, we excluded patients who met the exclusion criteria. Number of patients in full analysis set was 82 patients. Of which, 13 patients were positive for anti-THPO antibody, account for 15.9%. 69 patients without anti-THPO antibody in full analysis set were selected as controls.
2) How about the baseline characteristics?
Thank you for your comment. The clinical and laboratory characteristics of the patients according to the presence or absence of the anti-THPO antibody are summarized in Table 1. For further analysis of baseline characteristics, we have added information about chronic complications of diabetes in Table 1. There were no significant differences in age, sex, chronic complications of diabetes, parameters of glycemic control and drug therapy for diabetes between patients with the anti-THPO antibody and those without the anti-THPO antibody.
3) How to define confounding factors in multivariate linear regression?
Thank you for your comment. In table 3, we selected age, HbA1c, CAP, white blood cell count and MPV as covariates of multivariate linear regression analyses, because those factors were positively correlated with platelet count in the univariate correlation analyses.
When we re-analyzed the data using all factors, the same trends were observed as below.
|
Crude |
model 1 |
model 2 |
||||||
|
Characteristic |
r |
p |
|
β |
p |
|
β |
p |
|
Age (years) |
-0.35 |
<0.01 |
−0.27 |
<0.01 |
-0.22 |
0.11 |
||
|
Sex (male) |
0.09 |
0.41 |
- |
- |
-0.02 |
0.83 |
||
|
Duration of diabetes (years) |
-0.09 |
0.42 |
- |
- |
-0.27 |
<0.05 |
||
|
Hemoglobin A1c (%) |
0.32 |
<0.01 |
0.25 |
<0.05 |
0.38 |
<0.01 |
||
|
eGFR (ml/min/1.73 m2) |
0.21 |
0.06 |
- |
- |
0.12 |
0.30 |
||
|
C-reactive protein (mg/dL) |
0.15 |
0.19 |
- |
- |
0.03 |
0.76 |
||
|
Liver stiffness measurement (kPa) |
0.15 |
0.17 |
- |
- |
0.11 |
0.30 |
||
|
Controlled attenuation parameter (dB/m) |
0.25 |
<0.05 |
0.12 |
0.23 |
0.19 |
0.07 |
||
|
White blood cell (×109/l) |
0.24 |
<0.05 |
0.14 |
0.18 |
0.12 |
0.27 |
||
|
Hemoglobin (g/dL) |
0.03 |
0.76 |
- |
- |
-0.21 |
0.08 |
||
|
Mean platelet volume (fL) |
-0.25 |
<0.05 |
−0.34 |
<0.01 |
-0.29 |
<0.01 |
||
|
Thrombopoietin (pg/mL) |
0.02 |
0.84 |
- |
- |
0.14 |
0.20 |
||
|
anti-THPO antibody (yes) |
-0.29 |
<0.01 |
|
−0.23 |
<0.05 |
|
-0.24 |
<0.05 |
Reviewer 2 Report
The aim of the manuscript submitted by Fukuda and colleagues is to investigate the prevalence of anti-THPO antibody in patients affected by T2D. The results presented are consistent with the aim of the study and they are novel and of interest concerning diabetes physiopathology. However, major significant issues arose reading the manuscript and for this reason, the manuscript needs major revision prior to a publication:
Major revisions:
- The author presented the study as a pilot one and they acknowledged the lack of a control group in the limitations. However, it is paramount to understand the prevalence of these autoantibodies also in the general population without diabetes and compare the results with T2D. The result of this analysis could totally change the result’s interpretation and relevance.
- No information about chronic complications of diabetes is reported in the manuscript. In order to investigate the pathophysiological role and clinical relevance of anti-THPO antibody could be interesting evaluate if there is any correlation between anti-THPO antibody presence and any complications of diabetes or any difference regarding complications prevalence in patients with and without anti-THPO antibody.
- In order to strengthen the findings of this study, it could be useful to show information (and potential discussion) on the sensibility and specificity of the ELISA test used to detect the anti-THPO antibody. If there are any in your knowledge, It could be useful also highlight potential bias or limitation in the test/evaluation of the anti-THPO antibody.
Minor revisions:
- It is well known that some patients diagnosed as T2D are affected by Latent autoimmune Diabetes of Adulthood (LADA) (you could refer to Nat Rev Endocrinol.2017 Nov;13(11):674-686). In LADA patients the prevalence of other autoimmune diseases is higher compared to T2D. It is possible to rule out or confirm this diagnosis in patients with the anti-THPO antibody.
- The results showed that the FIB4 index is significantly increased in patients with anti-THPO antibody, driven by low platelet counts. In order to give more insights to the reader, could you expand in the discussion the clinical relevance of this findings, how we can use in clinical practice this information and how anti-THPO antibody could impact on this parameter in clinical practice?
- Table 1: please, in order to make easier the reading of the results, could be useful provide the number and the percentage n(%) for all the binomial variables (yes/no), like gender, drugs and fatty liver.
Author Response
To Reviewer 2:
Major revisions:
1 The author presented the study as a pilot one and they acknowledged the lack of a control group in the limitations. However, it is paramount to understand the prevalence of these autoantibodies also in the general population without diabetes and compare the results with T2D. The result of this analysis could totally change the result’s interpretation and relevance.
Thank you for your insightful comment. We agree with you. Unfortunately, this study was based on a cohort study of diabetic patients, therefore we could not compare the positive rate of anti-THPO antibodies in diabetic patients with that in the general population without diabetes.
2 No information about chronic complications of diabetes is reported in the manuscript. In order to investigate the pathophysiological role and clinical relevance of anti-THPO antibody could be interesting evaluate if there is any correlation between anti-THPO antibody presence and any complications of diabetes or any difference regarding complications prevalence in patients with and without anti-THPO antibody.
Thank you for your kind suggestion. According to your suggestion, we have added information about chronic complications of diabetes in Table 1. We also think that it is very interesting to evaluate if there is any correlation between anti-THPO antibody presence and any complications of diabetes or any difference regarding complications prevalence in patients with and without anti-THPO antibody. However, the parameters of glycemic control and chronic complications of diabetes were almost the same between the two groups. Then we have added the sentences in the Results section as below (Page 2, line 81):
“The parameters of glycemic control and chronic complications of diabetes were almost the same between the two groups.”
3 In order to strengthen the findings of this study, it could be useful to show information (and potential discussion) on the sensibility and specificity of the ELISA test used to detect the anti-THPO antibody. If there are any in your knowledge, It could be useful also highlight potential bias or limitation in the test/evaluation of the anti-THPO antibody.
Thank you for your kind suggestion. We agree with you. Unfortunately, however, anti-THPO antibody ELISA used in this study was a made-to-order product, and the sensitivity and specificity data cannot be shown at this time. In addition to that, it is difficult for us to conduct another assay as a gold standard for the diagnosis of anti-THPO antibody induced thrombocytopenia.
Minor revisions:
1 It is well known that some patients diagnosed as T2D are affected by Latent autoimmune Diabetes of Adulthood (LADA) (you could refer to Nat Rev Endocrinol.2017 Nov;13(11):674-686). In LADA patients the prevalence of other autoimmune diseases is higher compared to T2D. It is possible to rule out or confirm this diagnosis in patients with the anti-THPO antibody.
Thank you for your comment. Patients enrolled in the cohort study were routinely checked for anti-GAD antibodies at the first visit to our hospital. We excluded 9 patients who were positive for anti-GAD antibodies, as shown in Figure 1. Therefore, it is considered that LADA patients were not included in this study group.
2 The results showed that the FIB4 index is significantly increased in patients with anti-THPO antibody, driven by low platelet counts. In order to give more insights to the reader, could you expand in the discussion the clinical relevance of this findings, how we can use in clinical practice this information and how anti-THPO antibody could impact on this parameter in clinical practice?
We thank you for your careful reading of the manuscript and helpful comment.
According to your suggestion, we have added the sentences in the Discussion section as below (Page 6, line 128-131):
“Furthermore, when we see patients with low platelets and no findings of suggesting nonalcoholic steatohepatitis (NASH) or cirrhosis, the possibility of anti-THPO antibody needs to be considered, in addition to drug-related thrombocytopenia or ITP.”
3 Table 1: please, in order to make easier the reading of the results, could be useful provide the number and the percentage n(%) for all the binomial variables (yes/no), like gender, drugs and fatty liver.
Thank you for your precise suggestion.
According to your suggestion, we have modified Table 1.
|
Characteristics |
Anti-THPO antibody (−) |
Anti-THPO antibody (+) |
p |
|
Total number |
69 |
13 |
- |
|
Male |
30 (43.5) |
5 (38.5) |
0.74 |
|
Age (years) |
64.1 ± 11.3 |
69.9 ± 10.2 |
0.09 |
|
Duration of diabetes (years) |
13.9 ± 9.8 |
9.6 ± 7.7 |
0.15 |
|
Body weight (kg) |
63.6 ± 13.7 |
57.3 ± 14.3 |
0.14 |
|
Body mass index (kg/m2) |
24.3 ± 3.8 |
23.7 ± 4.1 |
0.56 |
|
Smoking (none / past / current) |
40 (58.0)/ 23 (33.3)/ 6 (8.7) |
4 (30.8) / 8 (61.5) / 1 (7.7) |
0.15 |
|
Consumption of alcohol (g/day) |
13.8 ± 4.7 |
8.8 ± 8.8 |
0.67 |
|
Retinopathy (NDR/SDR/PDR) |
51 (73.9)/13 (18.8)/5 (7.3) |
10 (76.9)/1 (7.7)/2 (1.4) |
0.44 |
|
Nephropathy (normo/micro/macroalbumiuria) |
45 (65.2)/19 (27.5)/5 (7,3) |
10 (76.9)/3 (23.1)/0 (0) |
0.74 |
|
Neuropathy (no/yes) |
44 (63.8)/25 (36.2) |
6 (46.2)/ 7 (53.8) |
0.23 |
|
Sulfonylurea |
26 (37.7) |
6 (46.2) |
0.57 |
|
Dipeptidyl peptidase-4 inhibitor |
16 (23.2) |
5 (38.5) |
0.26 |
|
Glucagon like peptide-1 receptor agonist |
4 (5.8) |
0 (0) |
0.23 |
|
Pioglitazone |
23 (33.3) |
6 (46.2) |
0.38 |
|
Metformin |
32 (46.4) |
4 (30.8) |
0.29 |
|
SGLT2 inhibitor |
2 (2.9) |
0 (0) |
0.40 |
|
Insulin therapy |
22 (31.9) |
4 (30.8) |
0.94 |
|
Systolic blood pressure (mmHg) |
130.1 ± 17.3 |
137.8 ± 22.3 |
0.16 |
|
Diastolic blood pressure (mmHg) |
76.2 ± 11.8 |
81.0 ± 11.8 |
0.18 |
|
LSM (kPa) |
4.63 ± 1.13 |
5.08 ± 1.29 |
0.19 |
|
CAP (dB/m) |
262.1 ± 55.0 |
247.8 ± 55.9 |
0.39 |
|
Fatty liver |
47 (68.1) |
7 (53.8) |
0.33 |
|
FIB4 index |
1.51 ± 0.68 |
1.92 ± 0.58 |
<0.05 |
|
HbA1c (%) |
7.43 ± 1.04 |
7.44 ± 1.25 |
0.97 |
|
Fasting plasma glucose (mg/dL) |
138.5 ± 35.5 |
144.6 ± 37.6 |
0.57 |
|
Aspartate aminotransferase (IU/L) |
23.2 ± 9.4 |
25.2 ± 9.2 |
0.49 |
|
Alanine aminotransferase (IU/L) |
23.9 ± 16.5 |
27.0 ± 16.7 |
0.53 |
|
Gamma-glutamyl transferase (IU/L) |
32.8 ± 23.1 |
30.5 ± 17.3 |
0.73 |
|
eGFR (ml/min/1.73 m2) |
76.6 ± 20.0 |
73.9 ± 19.1 |
0.66 |
|
C-reactive protein (mg/dL) |
0.05 ± 0.03 |
0.14 ± 0.21 |
0.11 |
|
Thrombopoietin (pg/mL) |
67.4 ± 63.0 |
91.1 ± 56.9 |
0.23 |
|
White blood cell count (×109/L) |
7.07 ± 1.83 |
5.60 ± 1.36 |
<0.01 |
|
Red blood cell count (×1012/L) |
4.57 ± 0.50 |
4.73 ± 0.56 |
0.31 |
|
Hemoglobin (g/dL) |
13.6 ± 1.5 |
14.1 ± 1.4 |
0.31 |
|
MPV (fL) |
9.99 ± 0.91 |
9.68 ± 0.77 |
0.26 |
|
Platelets (×109/L) |
228.7 ± 53.3 |
186.7 ± 37.1 |
<0.01 |
Data are presented as n (%) or as mean ± SD.
CAP, controlled attenuation parameter; eGFR, estimated glomerular filtration rate; FIB4, fibrosis 4; HbA1c, hemoglobin A1c; LSM, liver stiffness measurement; MPV, mean platelet volume; NDR, no diabetic retinopathy; PDR, proliferative diabetic retinopathy; SDR, simple diabetic retinopathy; SGLT2, sodium-glucose cotransporter 2.
Round 2
Reviewer 2 Report
.